# Stoichiometry Representation Learning with Polymorphic Crystal Structures

**Namkyeong Lee**[1,3†], **Heewoong Noh**[1], **Gyoung S. Na**[2],
**Tianfan Fu**[3], **Jimeng Sun**[3], **Chanyoung Park**[1*]
[1] KAIST    [2] KRICT    [3] UIUC
{namkyeong96,heewoongnoh,cy.park}@kaist.ac.kr
ngs0@krict.re.kr, futianfan@gmail.com, jimeng@illinois.edu

## Abstract

Despite the recent success of machine learning (ML) in materials science, its success heavily relies on the structural description of crystal, which is itself computationally demanding and occasionally unattainable. Stoichiometry descriptors can be an alternative approach, which reveals the ratio between elements involved to form a certain compound without any structural information. However, it is not trivial to learn the representations of stoichiometry due to the nature of materials science called *polymorphism*, i.e., *a single stoichiometry can exist in multiple structural forms due to the flexibility of atomic arrangements*, inducing uncertainties in representation. To this end, we propose PolySRL, which learns the probabilistic representation of stoichiometry by utilizing the readily available structural information, whose uncertainty reveals the polymorphic structures of stoichiometry. Extensive experiments on sixteen datasets demonstrate the superiority of PolySRL, and analysis of uncertainties shed light on the applicability of PolySRL in real-world material discovery. The source code for PolySRL is available at https://github.com/Namkyeong/PolySRL_AI4Science.

## 1 Introduction

Recently, ML techniques have found their applications in the field of materials science to analyze the extensive amount of experimental and computational data available [65, 57]. However, the effectiveness of these ML models is not only influenced by the selection of appropriate models but also reliant on the *numerical descriptors* used to characterize the systems of interest. Although it is still an open problem to construct appropriate descriptions of materials, there is a general agreement on effective descriptors that encompass the following principles [22, 17, 2, 55, 40]: Descriptors should **1)** preserve the similarity or difference between two data points (*preservativity*), **2)** be applicable to the entire materials domain of interest (*versatility*), and **3)** be computationally more feasible to generate compared to computing the target property itself (*computability*).

Among various types of descriptors, there has been a notable surge of interest in using descriptors based on the knowledge of crystal structure in materials science. In particular, as shown in Figure 1(a), one can create graphical descriptions of crystalline systems by considering periodic boundary conditions and defining edges as connections between neighboring atoms within a specific distance [59, 10]. However, these graphical descriptors depend on the structural details of crystals, which are usually obtained through computationally demanding and, in some cases, infeasible Density Functional Theory (DFT) calculations [48]. As a result, graphical descriptors are limited by the same computational bottleneck as DFT calculations, violating the principles of versatility and computability [15].

---

[*]Corresponding author.    [†] Work done while the author was a visiting Ph.D. student in UIUC.

NeurIPS 2023 Workshop on AI for Scientific Discovery: From Theory to Practice (AI4Science).

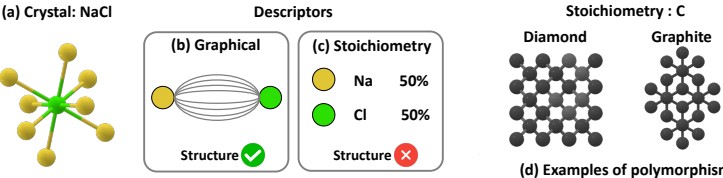

Figure 1: (a) Crystal structure of NaCl. (b), (c) Graphical and stoichiometry description of NaCl, respectively. (d) Diamond and Graphite share a single stoichiometry but have different structures.

An alternative approach to using graphical descriptors is to develop material representations from stoichiometry[2] alone, as shown in Figure 1(b), which generates the representation of material solely based on its elemental composition [25, 20]. Despite its simplicity, stoichiometry-based models have been shown to robustly offer a promising set of favorable elemental compositions for exploring new materials with cheap computational cost [15]. However, this approach is inherently limited in that it overlooks the structural information of crystals, leading to inferior performance compared to graphical models [1] given that structural details strongly influence the crystal properties. This naturally prompts a question: "Is it possible for stoichiometry-based models to also capture the structural information of crystals?"

To answer the question, we propose a novel multi-modal representation learning framework for stoichiometry that incorporates readily available crystal structural information (i.e., stoichiometry and crystal structural information as multi-modal inputs), inspired by the recent success of multi-modal contrastive learning approaches in various domains [18, 67]. For example, in computer vision, CLIP [46] improves the zero-shot transferability of a vision model by matching captions and images. Moreover, 3D Infomax [49] improves 2D molecular graph representation in quantum chemistry by maximizing the mutual information with its corresponding 3D molecular representations.

However, naively adopting existing multi-modal contrastive learning approaches to the stoichiometry representation learning task is non-trivial due to the intrinsic characteristics of crystal structures, i.e., one-to-many relationship between stoichiometry and crystal structures stemming from the flexibility of atomic arrangements, which is also known as *polymorphism*. In other words, solely relying on stoichiometry would contradict the principle of *preservativity*, especially for polymorphic materials with the same stoichiometry. More specifically, polymorphism refers to the nature of a certain compound to exist in different crystallographic structures due to different arrangements of atoms, resulting in totally different physical, and chemical properties [4]. An illustrative example of polymorphism is seen in the distinct forms of carbon: diamond and graphite (See Figure 1(c)). Diamond has a tetrahedral lattice structure with each carbon atom bonded to four others, resulting in its exceptional hardness and optical properties [9, 31]. However, graphite has a planar layered structure where carbon atoms are bonded in hexagonal rings, forming sheets that can easily slide past each other, giving graphite its lubricating and conducting properties [58, 29]. Therefore, it is essential not only to obtain qualified stoichiometry representations, but also to account for the uncertainties stemming from polymorphism for real-world material discovery, which has been overlooked in previous studies [20, 56].

To this end, we propose Polymorphic Stoichiometry Representation Learning (PolySRL), which aims to learn the representation of stoichiometry as a probabilistic distribution of polymorphs instead of a single deterministic representation [43, 14]. In particular, by assuming that polymorphs with an identical stoichiometry follow the same Gaussian distribution, PolySRL models each stoichiometry as a parameterized Gaussian distribution with learnable mean and variance vectors, whose distribution is trained to cover the range of polymorphic structures in representation space. By doing so, we expect the mean of Gaussian distribution serves as the representation of the stoichiometry, and the variance reflects the uncertainty stemming from the existence of various polymorphic structures, enabling PolySRL to assess the degree to which the representation adheres to the principle of *preservativity*. In this work, we make the following contributions:

- Recognizing the advantages and limitations of both structural and stoichiometry descriptors, we propose a multi-modal representation learning framework for stoichiometry, called PolySRL, which incorporates structural information of crystals into stoichiometry representations.

---

[2]Stoichiometry refers to the ratio between elements involved in a chemical reaction to form a compound.

- To capture uncertainties of stoichiometry stemming from various *polymorphs*, PolySRL learns a probabilistic representation for each stoichiometry instead of a deterministic representation.

- Extensive experiments on **sixteen datasets** demonstrate the superiority of PolySRL in learning representation of stoichiometry and predicting its physical properties. Moreover, we observe that measured uncertainties reflect various challenges in materials science, highlighting the applicability of PolySRL for real-world material discovery.

To the best of our knowledge, this is the first work that learns generalized representations of stoichiometry by simultaneously considering the crystal structural information and the polymorphism as uncertainty, which is crucial for the process of real-world material discovery.

## 2 Related Works

### 2.1 Graph Neural Networks for Materials

Among various ML methods, graph neural networks (GNNs) have been rapidly adopted by modeling crystal structures as graphical descriptions inspired by the recent success of GNNs in biochemistry [19, 50, 27, 21, 38, 37]. Specifically, CGCNN [59] first proposes a message-passing framework based on a multi-edge graph to capture interactions across cell boundaries, resulting in highly accurate prediction for eight distinct material properties. Building upon this multi-edge graph foundation, MEGNet [10] predicts various crystal properties by incorporating a physically intuitive strategy to unify multiple GNN models. Moreover, ALIGNN [12] proposes to utilize a line graph, in addition to a multi-edge graph, to model additional structural features such as bond angles and local geometric distortions. Despite the recent success of graph-based approaches, their major restriction is the requirement of atomic positions, which are typically determined through computationally intensive and sometimes infeasible DFT calculations. As a result, their effectiveness is mainly demonstrated in predicting properties for systems that have already undergone significant computational effort, restricting their utility in the materials discovery workflow [15].

### 2.2 Stoichiometry Representation Learning

Material representations can be alternatively constructed solely based on stoichiometry, which indicates the concentration of the constituent elements, without any knowledge of the crystal structure [15]. While stoichiometry has historically played a role in effective materials design [6, 45], it has been recently demonstrated that deep neural networks (DNNs) tend to outperform conventional approaches when large datasets are available. Specifically, ElemNet [25] takes elemental compositions as inputs and trains DNNs with extensive high-throughput OQMD dataset [36], showing improvements in performance as the network depth increases, up to a point where it reaches 17 layers. Roost [20] utilizes GNNs for stoichiometry representation learning by creating a fully connected graph in which nodes represent elements, allowing for the modeling of interactions between these elements. Instead of the message-passing scheme, CrabNet [56] introduces a self-attention mechanism to adaptively learn the representation of individual elements based on their chemical environment. While these methods are trained for a specific task, PolySRL aims to learn generalized stoichiometry representations for various tasks considering 1) the structural information and 2) polymorphism in crystal, both of which have not been explored before.

### 2.3 Probabilistic Representation Learning

First appearing in 2014 with the introduction of probabilistic word embeddings [53], probabilistic representations got a surge of interest from ML researchers by offering numerous benefits in modeling uncertainty pertaining to a representation. Specifically, in the computer vision domain, Shi & Jain [47] proposes to probabilistically represent face images to address feature ambiguity in real-world face recognition. Moreover, Oh et al. [43] introduces Hedged Instance Embeddings (HIB), which computes a match probability between point estimates but integrates it over the predicted distributions via Monte Carlo estimation. This idea has been successfully extended to cross-modal retrieval [14], video representation learning [44], and concept prediction [32]. In this paper, we aim to learn a probabilistic representation of stoichiometry, where the uncertainties account for various polymorphs associated with a single stoichiometry, enhancing the reliability of the material discovery process.

# 3 Preliminaries

## 3.1 Stoichiometry Graph Construction

Given a stoichiometry, we use $\mathcal{E} = \{e_1, \ldots, e_{n_e}\}$ to denote its unique set of elements, and $\mathcal{R} = \{r_1, \ldots, r_{n_e}\}$ to denote the compositional ratio of each element in the stoichiometry. We construct a fully connected stoichiometry graph $\mathcal{G}^a = (\mathcal{E}, \mathcal{R}, \mathbf{A}^a)$, where $\mathbf{A}^a \in \{1\}^{n_e \times n_e}$ indicates the adjacency matrix of a fully connected graph [20]. Then, we adopt GNNs as the stoichiometry encoder $f^a$, which aims to learn the stoichiometry representation by capturing complex relationships between elements via the message-passing scheme. Additionally, $\mathcal{G}^a$ is associated with an elemental feature matrix $\mathbf{X}^a \in \mathbb{R}^{n_e \times F}$ where $F$ is the number of features.

## 3.2 Structural Graph Construction

Given a crystal structure $(\mathbf{P}, \mathbf{L})$, suppose the unit cell has $n_s$ atoms, we have $\mathbf{P} = [\mathbf{p}_1, \mathbf{p}_2, \ldots, \mathbf{p}_{n_s}]^\mathsf{T} \in \mathbb{R}^{n_s \times 3}$ indicating the atom position matrix and $\mathbf{L} = [\mathbf{l}_1, \mathbf{l}_2, \mathbf{l}_3]^\mathsf{T} \in \mathbb{R}^{3 \times 3}$ representing the lattice parameter describing how a unit cell repeats itself in three directions. Based on the crystal parameters, we construct a multi-edge graph $\mathcal{G}^b = (\mathcal{V}, \mathbf{A}^b)$ that captures atom interactions across cell boundaries [59]. Specifically, $v_i \in \mathcal{V}$ denotes an atom $i$ and all its duplicates in the infinite 3D space whose positions are included in the set $\{\hat{\mathbf{p}}_i | \hat{\mathbf{p}}_i = \mathbf{p}_i + k_1 \mathbf{l}_1 + k_2 \mathbf{l}_2 + k_3 \mathbf{l}_3, k_1, k_2, k_3 \in \mathbb{Z}\}$, where $\mathbb{Z}$ denotes the set of all the integers. Moreover, $\mathbf{A}^b \in \{0, 1\}^{n_s \times n_s}$ denotes an adjacency matrix, where $\mathbf{A}^b_{i,j} = 1$ if two atoms $i$ and $j$ are within the predefined radius $r$ and $\mathbf{A}^b_{ij} = 0$ otherwise. Moreover, a single stoichiometry graph $\mathcal{G}^a$ is associated with a set of polymorphic crystal structural graphs $\mathcal{P}^{\mathcal{G}^a}$, i.e., $\mathcal{P}^{\mathcal{G}^a} = \{\mathcal{G}^b_1, \ldots, \mathcal{G}^b_{n_p}\}$, where $n_p$ is the number of polymorphs for the stoichiometry. Note that each node in $\mathcal{G}^b$ is associated with a learnable feature $\mathbf{x}^b \in \mathbb{R}^F$, which is shared across all crystals, to make sure we utilize only structural information. We provide further details on structural graph construction in Appendix A.

## 3.3 Task Descriptions

Given the stoichiometry graph $\mathcal{G}^a$ and the structural graph $\mathcal{G}^b$ of a single crystal, our objective is to acquire a stoichiometry encoder denoted as $f^a$, alongside mean and variance modules referred to as $f^a_\mu$ and $f^a_\sigma$, which associate structural information of $\mathcal{G}^b$ into latent representation of stoichiometry $\mathcal{G}^a$. Then, the modules are applied to a range of downstream tasks, a scenario frequently encountered in real-world material science where *solely stoichiometry of material is accessible*.

# 4 Methodology: PolySRL

In this section, we present Polymorphic Stoichiometry Representation Learning (PolySRL), which learns the representation of stoichiometry regarding polymorphic structures of crystals. Overall model architecture is illustrated in Figure 2.

## 4.1 Structural Graph Encoder

While structural information plays an important role in determining various properties of crystals, previous studies have overlooked the readily available crystal structures [23] for stoichiometry representation learning [25, 20, 56]. To this end, we use a GNN encoder to learn the representation of crystal structure, which is expected to provide guidance for learning the representation of stoichiometry. More formally, given the crystal structural graph $\mathcal{G}^b = (\mathbf{x}^b, \mathbf{A}^b)$, we obtain a structural representation of a crystal as follows:

$$\mathbf{z}^b = \text{Pooling}(\mathbf{Z}^b), \quad \mathbf{Z}^b = f^b(\mathbf{x}^b, \mathbf{A}^b), \tag{1}$$

where $\mathbf{Z}^b \in \mathbb{R}^{n_s \times F}$ is a matrix whose each row indicates the representation of each atom in the crystal structure, $\mathbf{z}^b$ indicates the latent representation of a crystal structure, and $f^b$ is the GNN-based crystal structural encoder. In this paper, we adopt graph networks [3] as the encoder, which is a generalized version of various GNNs, and sum pooling is used as the pooling function. We provide further details on the GNNs in Appendix B.1.

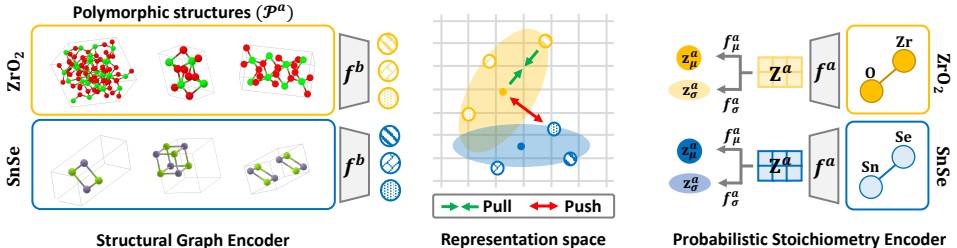

Figure 2: Overall model architecture.

## 4.2 Probabilistic Stoichiometry Encoder

**Deterministic Representation.** After obtaining the structural representation $\mathbf{z}^b$, we also compute the stoichiometry representation from the stoichiometry graph $\mathcal{G}^a$ as follows:

$$\mathbf{z}^a = \text{Pooling}(\mathbf{Z}^a), \quad \mathbf{Z}^a = f^a(\mathbf{X}^a, \mathbf{A}^a), \tag{2}$$

where $\mathbf{Z}^a \in \mathbb{R}^{n_e \times F}$ is a matrix whose each row indicates the representation of each element in a stoichiometry, $\mathbf{z}^a \in \mathbb{R}^F$ indicates the stoichiometry representation of a crystal, and $f^a$ is a GNN-based stoichiometry encoder. By utilizing GNNs, the stoichiometry encoder effectively learns intricate relationships and chemical environments related to elements, thereby enhancing the stoichiometry representation in a systematic manner [20]. For the stoichiometry encoder $f^a$, we adopt GCNs [35] with jumping knowledge [60], and weighted sum pooling with the compositional ratio (i.e., $\mathcal{R}$ in Section 3.1) is used as the pooling function.

One straightforward approach for injecting structural information into the stoichiometry representation would be adopting the idea of recent multi-modal contrastive learning approaches, which have been widely known to maximize the mutual information between heterogeneous modality inputs (two modalities in our case: stoichiometry and structure) [46, 49]. However, such a naive adoption fails to capture the polymorphic nature of crystallography: *A single stoichiometry can result in multiple distinct structures due to the diverse atomic arrangements, leading to significantly different physical, and chemical properties* [4]. That is, the relationship between the representations $\mathbf{z}^a$ and $\mathbf{z}^b$ constitutes a one-to-many mapping rather than a one-to-one mapping, leading to inherent uncertainties in the stoichiometry representation $\mathbf{z}^a$.

**Probabilistic Representation.** To this end, we propose to learn a probabilistic representation of stoichiometry $\mathbf{z}^a$, which naturally exhibits uncertainties of the representation, inspired by the recent Hedge Instance Embeddings (HIB) [43]. The main idea here is to learn the Gaussian representation of stoichiometry, which reveals the distribution of polymorphic structures $\mathcal{P}^a$ in representation space. Intuitively, the variance of this distribution reflects the range of diversity within these structures, giving us an idea of how well the representation adheres to the principle of *preservativity*. More formally, we model each stoichiometry as a parameterized Gaussian distribution with learnable mean vectors and diagonal covariance matrices as follows:

$$p(\tilde{\mathbf{z}}^a|\mathbf{X}^a, \mathbf{A}^a) \sim \mathcal{N}(\mathbf{z}^a_\mu, \mathbf{z}^a_\sigma), \quad \text{where} \quad \mathbf{z}^a_\mu = f^a_\mu(\mathbf{Z}^a), \quad \mathbf{z}^a_\sigma = f^a_\sigma(\mathbf{Z}^a). \tag{3}$$

Here, $\mathbf{z}^a_\mu, \mathbf{z}^a_\sigma \in \mathbb{R}^F$ denote the mean vector and the diagonal entries of the covariance matrix, respectively, and $f^a_\mu$ and $f^a_\sigma$ refer to the modules responsible for calculating the mean and diagonal covariance matrices, respectively. During training, we adopt the re-parameterization trick [34] to obtain samples from the distribution, i.e., $\tilde{\mathbf{z}}^a = diag(\sqrt{\mathbf{z}^a_\sigma}) \cdot \epsilon + \mathbf{z}^a_\mu$, where $\epsilon \sim \mathcal{N}(0, 1)$. While mean and variance are obtained from the shared $\mathbf{Z}^a$, we utilize different attention-based set2set pooling functions for $f^a_\mu$ and $f^a_\sigma$ [54], since the attentive aspects involved in calculating the mean and variance should be independent from each other. We provide further details on the probabilistic stoichiometry encoder in Appendix B.2.

## 4.3 Model Training via Representation Alignment

To incorporate the structural information into the stoichiometry representation, we define a matching probability between the stoichiometry graph $\mathcal{G}^a$ and its corresponding set of polymorphic crystal

structural graphs $\mathcal{P}^{\mathcal{G}^a}$ in the Euclidean space as follows:

$$p(m|\mathcal{G}^a, \mathcal{P}^{\mathcal{G}^a}) \approx \sum_{p \in \mathcal{P}^a} \frac{1}{J} \sum_{j=1}^{J} \text{sigmoid}\big(-c\|\tilde{\mathbf{z}}_j^a - \mathbf{z}_p^b\|_2 + d\big), \qquad (4)$$

where $\tilde{\mathbf{z}}_j^a$ is the sampled stoichiometry representation (Section 4.2), $\mathbf{z}_p^b$ is the structural graph representation (Section 4.1), $c, d > 0$ are parameters learned by the model for soft threshold in the Euclidean space, $J$ is the number of samples sampled from the distribution, and $\text{sigmoid}(\cdot)$ is the sigmoid function. Moreover, $m$ is the indicator function of value 1 if $\mathcal{P}^{\mathcal{G}^a}$ is the set of polymorphic structures corresponding to $\mathcal{G}^a$ and 0 otherwise. Then, we apply the soft contrastive loss [43, 14] as:

$$\mathcal{L}_{\text{con}} = \begin{cases} -\log p(m|\mathcal{G}^a, \mathcal{P}^{\mathcal{G}^{a'}}), & \text{if } a = a', \\ -\log\left(1 - p(m|\mathcal{G}^a, \mathcal{P}^{\mathcal{G}^{a'}})\right), & \text{otherwise.} \end{cases} \qquad (5)$$

Intuitively, the above loss aims to minimize the distance between a sampled stoichiometry representation and its associated polymorphic structural representations, while maximizing the distance between others. By doing so, PolySRL learns a probabilistic stoichiometry representation that considers the structural information and its associated uncertainties, which tend to increase when multiple structures are associated with a single stoichiometry, i.e., polymorphism.

In addition to the soft contrastive loss, we utilize a KL divergence loss between the learned stoichiometry distributions and the standard normal distribution $\mathcal{N}(0, 1)$, i.e., $\mathcal{L}_{\text{KL}} = \text{KL}(p(\tilde{\mathbf{z}}^a|\mathbf{X}^a, \mathbf{A}^a) \,\|\, \mathcal{N}(0, 1))$, which prevents the learned variances from collapsing to zero. Therefore, our final loss for model training is given as follows:

$$\mathcal{L}_{\text{total}} = \mathcal{L}_{\text{con}} + \beta \cdot \mathcal{L}_{\text{KL}}, \qquad (6)$$

where $\beta$ is the hyperparameter for controlling the weight of the KL divergence loss. During the inference, we use the mean vector $\mathbf{z}_\mu^a$ as the stoichiometry representation and the geometric mean of diagonal covariance matrices $\mathbf{z}_\sigma^a$ as uncertainty [14].

## 5 Experiments

### 5.1 Experimental Setup

**Datasets.** For training PolySRL, we collect 80,162 unique stoichiometries and their corresponding 112,183 DFT-calculated crystal structures from **Materials Project (MP)** website [3]. However, since DFT-calculated properties often deviate from real-world wet-lab experimental properties [26], we primarily evaluate PolySRL using wet-lab experimental datasets. Specifically, we use publicly available datasets containing experimental properties of stoichiometry, including **Band Gap** [66], **Formation Enthalpies** [33], **Metallic** [39], and **ESTM** [41]. Moreover, we conduct experiments on seven **Matbench** [16] datasets that are related to DFT-calculated properties. We provide further details on the datasets in Appendix C.

**Baseline Methods.** Since PolySRL is the first work that learns stoichiometry representation without any label information, we construct competitive baseline models from other domains. **Rand init.** refers to a randomly initialized stoichiometry encoder without any training process. **GraphCL** [62] learns the stoichiometry representation based on random augmentations on the stoichiometry graph $\mathcal{G}^a$, without utilizing structural information. **MP Band G.** and **MP Form. E.** learn the stoichiometry representation by predicting the DFT-calculated properties, which are available in **MP** database [3], i.e., band gap and formation energy per atom, respectively. **3D Infomax** [49] learns stoichiometry representation by maximizing the mutual information between stochiometry graph $\mathcal{G}^a$ and structural graph $\mathcal{G}^b$ with NTXent (Normalized Temperature-scaled Cross Entropy) loss [11]. We provide further details on baseline methods in Appendix D. In addition, we also compare PolySRL with supervised stoichiometry representation learning methods, i.e., **Roost** [20] and **CrabNet** [56] in Appendix F.2.

**Evaluation Protocol.** We first train all models in an unsupervised manner without any use of wet-lab experimental data. Then, we evaluate PolySRL in two evaluation schemes, i.e., representation learning and transfer learning. We further provide the detailed evaluation protocols in Appendix E.

---

[3] https://materialsproject.org/

Table 1: Representation learning performance (MAE) (Prop.: Property / Str.: Structure / Poly.: Polymorphism / Band G.: Band Gap / Form. E.: Formation Entalphies / E.C.: Electrical Conductivity / T.C.: Thermal Conductivity).

| Model | DFT | | Poly. | Band G. | Form. E. | Metallic | ESTM 300K | | | ESTM 600K | | | $Z\bar{T}$ | |
| | Prop. | Str. | | | | | E.C. | T.C. | Seebeck | E.C. | T.C. | Seebeck | 300K | 600K |
|---|---|---|---|---|---|---|---|---|---|---|---|---|---|---|
| Rand init. | ✗ | ✗ | ✗ | 0.439 (0.014) | 0.671 (0.066) | 0.211 (0.023) | 1.029 (0.119) | 0.225 (0.030) | 0.451 (0.031) | 0.714 (0.113) | 0.218 (0.024) | 0.437 (0.087) | 0.099 (0.017) | 0.261 (0.160) |
| GraphCL | ✗ | ✗ | ✗ | 0.437 (0.022) | 0.677 (0.030) | 0.212 (0.019) | 1.057 (0.115) | 0.229 (0.040) | 0.459 (0.044) | 0.695 (0.119) | 0.206 (0.027) | 0.440 (0.077) | 0.121 (0.027) | 0.211 (0.043) |
| MP Band G. | ✓ | ✗ | ✗ | **0.403** (0.011) | 0.690 (0.043) | 0.212 (0.028) | 1.008 (0.081) | 0.225 (0.026) | 0.443 (0.074) | 0.690 (0.085) | 0.217 (0.023) | 0.436 (0.075) | 0.129 (0.044) | 0.251 (0.161) |
| MP Form. E. | ✓ | ✗ | ✗ | 0.416 (0.017) | 0.619 (0.062) | 0.203 (0.022) | 1.121 (0.137) | 0.228 (0.024) | 0.441 (0.078) | 0.784 (0.078) | 0.220 (0.021) | 0.444 (0.091) | 0.093 (0.008) | 0.328 (0.075) |
| 3D Infomax | ✓ | ✓ | ✗ | 0.428 (0.015) | 0.654 (0.032) | 0.201 (0.032) | 0.969 (0.110) | 0.217 (0.040) | 0.432 (0.070) | 0.692 (0.102) | 0.212 (0.013) | 0.428 (0.076) | 0.105 (0.030) | 0.171 (0.023) |
| PolySRL | ✓ | ✓ | ✓ | 0.407 (0.013) | **0.592** (0.039) | **0.194** (0.017) | **0.912** (0.121) | **0.197** (0.020) | **0.388** (0.059) | **0.665** (0.126) | **0.189** (0.017) | **0.412** (0.043) | **0.070** (0.014) | **0.168** (0.021) |

## 5.2 Empirical Results

**Representation Learning.** In Table 1, we have the following observations: **1)** Comparing the baseline methods that take into account structural information (**Str.** ✓) with those that do not (**Str.** ✗), we find out that utilizing structural information generally learns more high-quality stoichiometry representations. This is consistent with the established knowledge in crystallography, which emphasizes that structural details, including crystal structure and symmetry, play a crucial role in determining a wide range of physical, chemical, and mechanical properties [4, 5]. **2)** Moreover, we observe PolySRL outperforms baseline methods that overlook polymorphism in their model design. This highlights the significance of our probabilistic approach, which not only offers insights into polymorphism-related uncertainties but also yields high-quality representations. **3)** On the other hand, we notice that utilizing DFT-calculated values contributes to the model's understanding of a specific target property (see **Prop.** ✓). For instance, when the model is trained with a DFT-calculated band gap (i.e., MP Band G.), it surpasses all other models when predicting experimental band gap values. This highlights that knowledge acquired from DFT-calculated properties can be applied to wet-lab experimental datasets. However, these representations are highly tailored to a particular target property, which restricts their generalizability for diverse tasks. We also provide empirical results on Matbench datasets that contain DFT-calculated properties in Appendix F.1 and transfer learning scenarios in Appendix F.2.

**Physical Validity of Predicted Properties.** To further verify the physical validity of predicted properties, we theoretically calculate the figure of merit $Z\bar{T}$ [4] of thermoelectrical materials with the predicted properties in ESTM datasets in Table 1. More specifically, given predicted electrical conductivity (E.C.) $\sigma$, thermal conductivity (T.C.) $\lambda$, Seebeck coefficient $S$, we can compute the figure of merit $Z\bar{T}$ as follows: $Z\bar{T} = \frac{S^2\sigma}{\lambda}\bar{T}$, where $\bar{T}$ indicates a conditioned temperature, i.e., 300 K and 600 K. In Table 1, we have following observations: **1)** Looking at the general model performance on ESTM datasets and $Z\bar{T}$, we find that performing well on ESTM datasets does not necessarily indicate the predictions are physically valid. **2)** In contrast, models that incorporate structural information tend to produce physically valid predictions in both ESTM datasets, underscoring the importance of the crystal structural information. **3)** Moreover, PolySRL consistently outperforms baseline methods, demonstrating that PolySRL not only learns accurate representations of stoichiometry but also ensures the physical validity of the predictions. We provide further analysis on the predicted $Z\bar{T}$, and high throughput screening results of thermoelectrical materials in Appendix F.3.

## 5.3 Uncertainty Analysis

**Number of Structures.** In this section, we examine how uncertainties vary according to the number of possible structures. To do so, we first collect all possible structures of stoichiometry in Band Gap dataset from MP database[3] and **Open Quantum Materials Database (OQMD)**[5]. Subsequently, we compute the average uncertainties for stoichiometry groups with the same number of possible structures. In Figure 3 (a), we have the following observations: **1)** In

---

[4]In thermoelectric materials, the figure of merit $Z\bar{T}$ plays a fundamental role in determining how effectively power can be generated and energy can be harvested across various applications [42].

[5]https://oqmd.org/

general, the uncertainty of stoichiometry that has polymorphic structures (# possible structures $\geq 2$) was higher than that of the stoichiometry with a single structure (# possible structures $= 1$), demonstrating that PolySRL learns uncertainties regarding polymorphic structures.
**2)** On the other hand, an increase in the number of possible structures in OQMD leads to an increase in the uncertainty, demonstrating that PolySRL learns uncertainties related to the diverse polymorphic structures. Note that this trend is mainly shown in the OQMD dataset due to the fact that OQMD encompasses not only realistic but also theoretically possible structures, indicating that PolySRL acquires knowledge of theoretical uncertainties in materials science. **3)** Furthermore, we notice high uncertainties when there are no potential structures available (i.e., when # possible structures $= 0$) in comparison to stoichiometry with a single possible structure, suggesting that uncertainty contains information about the computational feasibility of the structure.

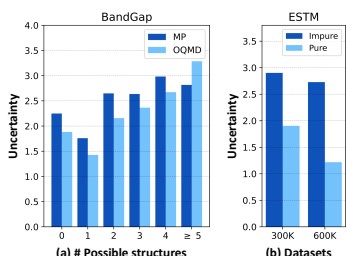

Figure 3: Uncertainty analysis.

**Impurity of Materials.** Next, we investigate how impurities in materials influence the uncertainty in stoichiometry. Specifically, we compare the average stoichiometry uncertainty between groups of doped or alloyed materials (i.e., Impure) and their counterparts (i.e., Pure) in thermoelectric materials datasets, i.e., ESTM 300K and ESTM 600K, where doping and alloying are commonly employed to enhance their performance. In Figure 3 (b), we notice a substantial increase in the uncertainty within impure materials compared with their pure counterparts. This observation is in line with common knowledge in materials science that doping or alloying can lead to chaotic transformations in a conventional structure [30, 28], demonstrating that PolySRL also captures the complexity of structure as the uncertainty. In conclusion, uncertainty analysis highlights that PolySRL effectively captures the uncertainty related to the presence of polymorphic structures within a single stoichiometry and the computational challenges associated with crystal structures.

**Case Studies.** While our previous analysis on uncertainties generally aligns with our expectations, we do observe some instances where PolySRL exhibits high uncertainty in non-polymorphic stoichiometries and minimal uncertainty in polymorphic stoichiometries. First, we observe the stoichiometry of HgCl and CaS exhibit high uncertainty, even though they only have one possible structure (Figure 4 (a)). We attribute this phenomenon to the limited availability of element combinations in the MP dataset, which occurred due to several factors, including the rarity of certain elements and the difficulty in synthesizing

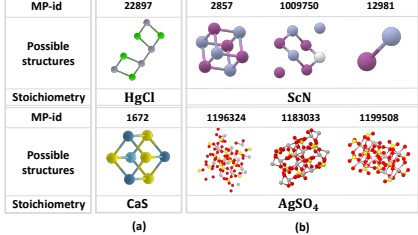

Figure 4: Case studies.

substances with specific combinations of elements [8, 24]. On the other hand, we observe the learned distribution of ScN and $AgSO_4$ collapsed to zero even though each of them has three possible polymorphic structures (Figure 4 (b)). This behavior arises from the structural similarity among the polymorphic structures, where all three polymorphic structures of each stoichiometry fall within the same cubic and monoclinic structural system, respectively. In conclusion, PolySRL acquires detailed insights concerning polymorphic structures beyond mere quantitative counts. Additionally, we include further analysis on the correlation between uncertainty and model performance, along with supplementary case studies that are in line with our anticipated results in Appendix F.5.

# 6 Conclusion

This paper focuses on learning a probabilistic representation of stoichiometry that incorporates polymorphic structural information of crystalline materials. Given stoichiometry and its corresponding polymorphic structures, PolySRL learns parameterized Gaussian distribution for each stoichiometry, whose mean becomes the representation of stoichiometry and variance indicates the level of uncertainty stemming from the polymorphic structures. Extensive empirical studies on sixteen datasets, including wet-lab experimental data and DFT-calculated data, have been conducted to validate the effectiveness of PolySRL in learning stoichiometry representations. Moreover, a comprehensive analysis of uncertainties reveals that the model learns diverse complexities encountered in materials science, highlighting the practicality of PolySRL in real-world material discovery process.

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

# Supplementary Material for
# Stoichiometry Representation Learning with Polymorphic Crystal Structures

This is an Appendix for the paper **Stoichiometry Representation Learning with Polymorphic Crystal Structures**, which is organized as follows: Section A provides details on constructing structural graph representation of crystalline materials. Section B elaborates on the implementation details of our method. Section C details all the datasets we use. Section D details the experimental setup of all the baseline methods. Section E describes evaluation protocol. Section F provides additional experimental results. Section G lists important notations used during the main manuscript.

## A    Structural Graph Construction

In this section, we provide the detailed structural graph construction process with a figure. Overall, this structural graph is the same as previous works [59, 61]. Given a crystal structure $(\mathbf{P}, \mathbf{L})$, suppose the unit cell has $n_s$ atoms, we have $\mathbf{P} = [\mathbf{p}_1, \mathbf{p}_2, \ldots, \mathbf{p}_{n_s}]^\mathsf{T} \in \mathbb{R}^{n_s \times 3}$ indicating the atom position matrix and $\mathbf{L} = [\mathbf{l}_1, \mathbf{l}_2, \mathbf{l}_3]^\mathsf{T} \in \mathbb{R}^{3 \times 3}$ representing the lattice parameter describing how a unit cell repeats itself in three directions. Since the crystal usually possesses irregular shapes in practice, $\mathbf{l}_1, \mathbf{l}_2, \mathbf{l}_3$ are not always orthogonal in 3D space [61]. For clear visualization, we provide examples of periodic patterns in 2D space in Figure 5 (a).

Based on the crystal parameters $(\mathbf{P}, \mathbf{L})$, we construct a multi-edge graph $\mathcal{G}^b = (\mathcal{V}, \mathbf{A}^b)$ that captures atom interactions across cell boundaries [59]. Specifically, $v_i \in \mathcal{V}$ denotes an atom $i$ and all its duplicates in the infinite 3D space whose positions are included in the set $\{\hat{\mathbf{p}}_i | \hat{\mathbf{p}}_i = \mathbf{p}_i + k_1 \mathbf{l}_1 + k_2 \mathbf{l}_2 + k_3 \mathbf{l}_3, k_1, k_2, k_3 \in \mathbb{Z}\}$, where $\mathbb{Z}$ denotes the set of all the integers. Moreover, $\mathbf{A}^b \in \{0, 1\}^{n_s \times n_s}$ denotes an adjacency matrix, where $\mathbf{A}_{i,j}^b = 1$ if two atoms $i$ and $j$ are within the predefined radius $r$ and $\mathbf{A}_{ij}^b = 0$ otherwise. Specifically, nodes $v_i$ and $v_j$ are connected if there exists any combination $k_1, k_2, k_3 \in \mathbb{Z}$ such that the euclidean distance $d_{ij}$ satisfies $d_{ij} = \|\mathbf{p}_i + k_1 \mathbf{l}_1 + k_2 \mathbf{l}_2 + k_3 \mathbf{l}_3 - \mathbf{p}_j\|_2 \le r$ (see Figure 5 (b)). For the initial feature for edges, we expand the distance $d_{ij}$ between atom $v_i$ and $v_j$ by Gaussian basis following previous works [59]. Moreover, for each node in $\mathcal{G}^b$ is associated with a learnable feature $\mathbf{x}^b \in \mathbb{R}^F$, which is shared across all crystals, to make sure we utilize only structural information.

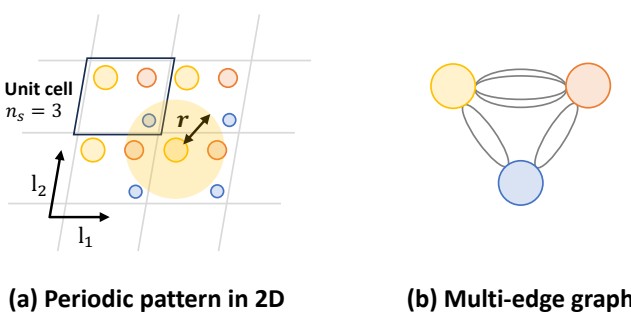

(a) Periodic pattern in 2D          (b) Multi-edge graph

Figure 5: Structural graph construction.

## B    Implementation Details

In this section, we provide implementation details of PolySRL.

### B.1    Structural Graph Encoder

Our structural graph encoder comprises two components: the encoder and the processor. The encoder acquires the initial representation of atoms and bonds, while the processor is responsible for learning how to pass messages throughout the crystal structure. More formally, given an atom $v_i$ and the bond $e_{ij}$ between atom $v_i$ and $v_j$ in crystal structure, node encoder $\phi_{node}$ and edge encoder $\phi_{edge}$ outputs initial representations of atom $v_i$ and bond $e_{ij}$ as follows:

$$\mathbf{h}_i^{0,b} = \phi_{node}(\mathbf{X}^b), \quad \mathbf{b}_{ij}^{0,b} = \phi_{edge}(\mathbf{B}_{ij}^b), \tag{7}$$

where $\mathbf{X}^b \in \mathbb{R}^{n_s \times F}$ is the atom feature matrix whose $i$-th row indicates the input feature of atom $v_i$, $\mathbf{B}^b \in \mathbb{R}^{n_s \times n_s \times F}$ is the bond feature tensor. As previously explained in Section 3.2, we employ

a common $\mathbf{x}^b$ for all atoms across all crystals, resulting in every row in $\mathbf{X}^b$ being identical to $\mathbf{x}^b$. With the initial representations of atoms and bonds, the processor learns to pass messages across the crystal structure and update atom and bond representations as follows:

$$\mathbf{b}_{ij}^{l+1,b} = \psi_{edge}^l(\mathbf{h}_i^{l,b}, \mathbf{h}_j^{l,b}, \mathbf{b}_{ij}^{l,b}), \quad \mathbf{h}_i^{l+1,b} = \psi_{node}^l(\mathbf{h}_i^{l,b}, \sum_{j \in \mathcal{N}(i)} \mathbf{b}_{ij}^{l+1,b}), \qquad (8)$$

where $\mathcal{N}(i)$ is the neighboring atoms of atom $v_i$, $\psi$ is a two-layer MLP with non-linearity, and $l = 0, \ldots, L'$. Note that $\mathbf{h}_i^{L',b}$ is equivalent to the $i$-th row of the atom embedding matrix $\mathbf{Z}^b$ in Equation 1. In this paper, we use a 3-layered structural graph encoder, i.e., $L' = 3$.

### B.2 Probabilistic Stoichiometry Encoder

**Stoichiometry Graph Encoder** $f^a$**.** For the stoichiometry graph encoder $f^a$, we utilize the architecture of GCNs [35] and Jumping Knowledge Network [60]. Specifically, given elemental feature matrix $\mathbf{X}^a$ and adjacency $\mathbf{A}^a$, GCN layers pass the messages to obtain latent elemental feature matrix as follows:

$$\mathbf{h}_i^{l+1,a} = \text{GCN}^l(\mathbf{h}_i^{l,a}, \mathbf{A}^a), \qquad (9)$$

where $\mathbf{h}_i^{0,a}$ indicates $i$-th row of elemental feature matrix $\mathbf{X}^a$, and $l = 0, \ldots, L'$. After $L'$ step message passing steps, we obtain a final representation of stoichiometry as follows:

$$\mathbf{Z}_i^a = \mathbf{W}(\text{Concat}[\mathbf{h}_i^{0,a}, \cdots, \mathbf{h}_i^{L',a}]), \qquad (10)$$

where $\mathbf{W} \in \mathbb{R}^{F \times L'F}$ is a learnable weight matrix that reduces the dimension of concatenated representations. Note that $\mathbf{Z}_i^a$ is equivalent to the $i$-th row of the element embedding matrix $\mathbf{Z}^a$ in Equation 2. We also use $L' = 3$ for stoichiometry encoder $f^a$. After obtaining the elemental representation matrix $\mathbf{Z}^a$, we obtain stoichiometry representation $\mathbf{z}^a$ by employing weighted sum pooling, which takes into account the compositional ratio.

**Mean** $f_\mu^a$ **and Variance** $f_\sigma^a$ **Module.** After obtaining the elemental representation matrix $\mathbf{Z}^a$, we utilize set2set [54] pooling function to obtain the mean vector and diagonal entries of the covariance vector. More specifically, given $\mathbf{Z}^a$, we obtain mean vector $\mathbf{z}_\mu^a$ and diagonal covariance vector $\mathbf{z}_\sigma^a$ as follows:

$$\mathbf{z}_\mu^a = \hat{\mathbf{z}}_\mu^a + \mathbf{z}^a, \quad \hat{\mathbf{z}}_\mu^a = \text{Set2set}_\mu(\mathbf{Z}^a), \qquad (11)$$

$$\mathbf{z}_\sigma^a = \hat{\mathbf{z}}_\sigma^a + \mathbf{z}^a, \quad \hat{\mathbf{z}}_\sigma^a = \text{Set2set}_\sigma(\mathbf{Z}^a). \qquad (12)$$

By obtaining mean and diagonal covariance vectors with separate pooling functions, i.e., Set2set$_\mu$ and Set2set$_\sigma$, the model learns different attentive aspects involved for each module.

### B.3 Training Details

We also describe the implementation details to enhance the reproducibility. Our method is implemented on Python 3.7.1, PyTorch 1.8.1, and Torch-geometric 1.7.0. All experiments are conducted using a 24GB NVIDIA GeForce RTX 3090. Model hyperparameters are given in Table 2. During training, we clip the gradient to the maximum value of 2 for stability [63].

Table 2: Hyperparameter specifications of PolySRL.

| # Layers | | Hidden | Learning | Batch | Epochs | Number of | $\beta$ | Initial | |
| --- | --- | --- | --- | --- | --- | --- | --- | --- | --- |
| $f^a$ | $f^b$ | dim ($F$) | Rate ($\eta$) | Size | | Samples ($J$) | | $c$ | $d$ |
| 3 | 3 | 200 | 5e-05 | 256 | 100 | 8 | 1e-08 | 20 | 20 |

## C  Datasets

In this section, we provide further details on the dataset used for experiments. We first introduce the datasets utilized for the main manuscript, which is mainly based on wet-lab experiments.

- **Materials Project** [23] is an openly accessible database that provides material properties calculated using density functional theory (DFT). We have gathered 80,162 distinct stoichiometries along with their corresponding 112,183 crystal structures computed using DFT, with up to 32,021 stoichiometries having multiple potential structures.

- **Band Gap** [66] dataset comprises experimentally determined band gap properties for non-metallic materials. It encompasses 2,482 distinct stoichiometries and a total of 3,895 experimental band gap values. Within this dataset, 1,413 instances of duplicate experimental band gap measurements for stoichiometries were identified. Consequently, our task involves predicting the band gap properties for these 2,482 stoichiometries, with the average value being computed in cases where duplicate experimental results exist for a given stoichiometry.

- **Formation Enthalpies** [33] dataset consists of experimentally determined formation enthalpy values for intermetallic phases and other inorganic compounds. It includes 1,141 unique stoichiometries and a total of 1,276 experimental formation enthalpy values. Within this dataset, 135 cases of duplicate experimental formation enthalpy measurements for stoichiometries were identified. Therefore, our objective is to predict the formation enthalpy properties for these 1,141 stoichiometries, calculating the average value when duplicate experimental results are present for a particular stoichiometry. We report MAE values multiplied by a factor of 10 for clear interpretation during all experiments.

- **Metallic** [39] dataset contains reduced glass transition temperature (Trg) for 584 unique metallic alloys. We report MAE values multiplied by a factor of 10 for clear interpretation during all experiments.

- **ESTM 300 K** [41] dataset contains various properties of 368 thermoelectric materials that are measured in the temperature range of 295 K to 305 K, which is widely recognized as room temperature in chemistry. Among the properties, we mainly target **electrical conductivity** ($S/m$), **thermal conductivity** ($W/mK$), and **Seebeck coefficient** ($\mu V/K$). Regarding electrical conductivity and thermal conductivity, we apply a logarithmic scaling to the target values because they exhibit significant skewness. Additionally, for the Seebeck coefficient, we use min-max scaling on the target values due to their wide range and report MAE values multiplied by a factor of 10 for clear interpretation during all experiments. When calculating the figure of merit ($Z\bar{T}$) with predicted properties, we reverse the scaling to return the original scale and then compute it.

- **ESTM 600 K** [41] dataset contains various properties of 188 thermoelectric materials that are measured in the temperature range of 593 K to 608 K, which is widely recognized as high temperature in chemistry. The properties we are targeting and the preprocessing steps applied are identical to those used for the **ESTM 300 K** dataset.

In addition to the wet-lab experimental datasets, we use following seven **Matbench** datasets that contain properties from DFT calculation.

- **Castelli Perovskites** [7] dataset contains formation energy of Perovskite cell of 18,928 materials.
- **Refractive Index** [23] dataset contains a refractive index of 4,764 materials, provided in **MP** database.
- **Shear Modulus** [23] dataset contains shear modulus of 10,987 materials, provided in **MP** database.
- **Bulk Modulus** [23] dataset contains bulk modulus of 10,987 materials, provided in **MP** database.
- **Exfoliation Energy** [13] dataset contains exfoliation energy 636 materials.
- **MP Band gap** [23] dataset contains band gap of 106,113 materials, provided in **MP** database.
- **MP Formation Energy** [23] dataset contains formation energy per atom in 132,752 materials, provided in **MP** database.

Following previous work [56, 20], we choose the target value associated with the lowest formation enthalpy for duplicate stoichiometries found in both the MP datasets, while we use the mean of the target values for other datasets.

## D Baseline Methods

In this section, we elaborate on baseline methods. For a fair comparison, all these baseline methods leverage the same neural network architecture and only differ in training objective function.

- **Rand init.** refers to the randomly initialized stoichiometry encoder without any training process.

- **GraphCL** [62] is a general graph-level contrastive learning strategy that uses random augmentation to construct positive and negative samples. In this paper, it learns the stoichiometry representation based on the random augmentation on the stoichiometry graph $\mathcal{G}^a$, without utilizing structural information. For the $n$-th data in the minibatch ($N$ data points), the loss function is defined as follows follows:

$$l_n = -\log \frac{\exp\{\text{sim}(z_n, z_n)/\tau\}}{\sum_{n'=1, n' \neq n}^{N} \exp\{\text{sim}(z_n, z_{n'})/\tau\}}, \tag{13}$$

where $\text{sim}(\cdot, \cdot)$ indicates cosine similarity between two latent vectors. $\tau > 0$ denotes temperature and is a hyperparameter. $z_i$ is the representation of the $i$-th data.

- **MP Band G.** and **MP Form. E.** learn the stoichiometry representation by predicting the DFT-calculated properties, i.e., band gap and formation energy per atom, respectively. More formally, model is trained with MAE loss for $n$-th data point in the minibatch ($N$ data points) as follows:

$$l_n = |Y_n - \hat{Y}_n|, \tag{14}$$

where $Y_n$ and $\hat{Y}_n$ denote DFT-calculated property and model prediction, respectively.

- **3D Infomax** [49] proposes to enhance model prediction on 3D molecular graphs by integrating 3D information of the molecules in its latent representations. Instead of 2D molecular graphs, we learn the representation of stoichiometry graph $\mathcal{G}^a$ by maximizing the mutual information with structural graph $\mathcal{G}^b$. More specifically, we train the model with NTXent (Normalized Temperature-scaled Cross Entropy) loss [11], which is defined for $n$-th data point in minibatch of size $N$ as follows:

$$l_n = -\log \frac{\exp\{\text{sim}(z_n^a, z_n^b)\}}{\sum_{n'=1, n' \neq n}^{N} \exp\{\text{sim}(z_n^a, z_{n'}^b)\}}, \tag{15}$$

where $\text{sim}(\cdot, \cdot)$ indicates cosine similarity between two latent vectors.

Even though the primary focus of this paper is to introduce training strategies for stoichiometry encoders without any label information, we also conduct a comparative analysis of our proposed approach with previous supervised stoichiometry representation learning methods [20, 56]. Note that these works propose sophisticated model architectures for stoichiometry representation learning, not training strategy.

- **Roost** [20] first proposes to utilize GNNs for stoichiometry representation learning by presenting stoichiometry as a fully connected graph, whose nodes are unique elements in stoichiometry. This approach allows the model to acquire distinct and material-specific representations for each element, enabling it to capture physically meaningful properties and interactions.

- **CrabNet** [56] designs a Transformer self-attention mechanism [51] to adaptively learn the representation of individual elements based on their chemical environment.

## E    Evaluation Protocol

**Evaluation Metrics.** We mainly compare the methods in terms of Mean Absolute Error (MAE) following previous work [20]. Moreover, we provide the model performance in terms of $R^2$ in Appendix F, which provides an intuitive measure of the fraction of the overall variance in the data that the model can account for.

During evaluation, we evaluate models in two different settings, i.e., representation learning and transfer learning. In both scenarios, we evaluate the model under a 5-fold cross-validation scheme, i.e., the dataset is randomly split into 5 subsets, and one of the subsets is used as the test set while the remaining subsets are used to train the model.

**Representation Learning.** For representation learning scenarios, we fix the model parameters (i.e., $f^a$, $f_\mu^a$, and $f_\sigma^a$) and train a three-layer MLP head with LeakyReLU non-linearity to evaluate the stoichiometry obtained by various models. Following previous works [52, 62], we train the MLP head with Adam optimizer with a fixed learning rate of 0.001 for 300 epochs.

**Transfer Learning.** For transfer learning scenarios, we allow the model parameters (i.e., $f^a$, $f_\mu^a$, and $f_\sigma^a$) to be trained with labels in downstream tasks, jointly with a three-layer MLP head

with LeakyReLU non-linearity. During the transfer learning stage, we train the model parameters and head with the Adam optimizer for 500 epochs. We tune the learning rate in the range of $\{0.005, 0.001, 0.0005, 0.0001\}$ with a validation set which is a subset $(20\%)$ of the training set. Due to the lack of data, we select the learning rate that yields the optimal performance on the validation set. Subsequently, we retrain the model using both the training set and the validation set, with the corresponding learning rate.

## F  Additional Experiments

### F.1  Experiments on DFT-Calculated Datasets

Although DFT-calculated properties frequently differ from actual wet-lab experimental properties [26], we have included experimental outcomes for seven DFT-calculated properties from the Matbench dataset [16]. These Matbench datasets were assessed using a five-fold cross-validation approach with train/validation/test splits set at a ratio of 72/8/20, as given in previous work [56]. In Table 3, we have following observations: **1)** In the DFT-based dataset, we observed significant disparities in trends compared to the experimental datasets in Table 1, demonstrating the inherent difference between the experimental data and DFT-calculated data. For instance, we noticed that the MP Form. E. model consistently outperforms the MP Band G. and 3D Infomax models. **2)** Furthermore, given that the datasets are designed to pick the target value linked to the lowest formation enthalpy among different polymorphic structures for a single stoichiometry, we find that models trained with specific DFT-calculated values (i.e., **Prop. ✓**) do not outperform models trained on corresponding datasets. This discrepancy is attributed to properties derived from non-lowest formation enthalpy polymorphic structures, which can introduce confusion to the model. **3)** However, we observe PolySRL generally outperforms baseline models, demonstrating its effectiveness in not only wet-lab experimental datasets but also in DFT-calculated datasets.

Table 3: Representation learning performance on DFT-calculated datasets (MAE).

| Model | DFT | | Poly. | Castelli Perovskites | Refractive Index | Shear Modulus | Bulk Modulus | Exfoliation Energy | MP | |
| | Prop. | Str. | | | | | | | Band G. | Form. E. |
|---|---|---|---|---|---|---|---|---|---|---|
| Rand init. | ✗ | ✗ | ✗ | 0.140 | 0.394 | 0.115 | 0.850 | 0.393 | 0.354 | 0.119 |
| | | | | (0.004) | (0.091) | (0.003) | (0.030) | (0.044) | (0.005) | (0.002) |
| GraphCL | ✗ | ✗ | ✗ | 0.145 | 0.386 | 0.117 | 0.844 | 0.411 | 0.351 | 0.121 |
| | | | | (0.006) | (0.094) | (0.002) | (0.021) | (0.060) | (0.004) | (0.001) |
| MP Band G. | ✓ | ✗ | ✗ | 0.141 | 0.399 | 0.116 | 0.851 | 0.397 | 0.354 | 0.119 |
| | | | | (0.004) | (0.085) | (0.002) | (0.042) | (0.041) | (0.007) | (0.002) |
| MP Form. E. | ✓ | ✗ | ✗ | 0.134 | **0.379** | 0.108 | **0.801** | 0.382 | 0.338 | 0.115 |
| | | | | (0.004) | (0.093) | (0.002) | (0.029) | (0.037) | (0.002) | (0.001) |
| 3D Infomax | ✓ | ✓ | ✗ | 0.147 | 0.388 | 0.117 | 0.880 | 0.408 | 0.354 | 0.116 |
| | | | | (0.004) | (0.094) | (0.003) | (0.040) | (0.043) | (0.005) | (0.002) |
| PolySRL | ✓ | ✓ | ✓ | **0.132** | 0.394 | **0.107** | 0.837 | **0.378** | **0.328** | **0.112** |
| | | | | (0.007) | (0.092) | (0.002) | (0.033) | (0.021) | (0.005) | (0.003) |

### F.2  Transfer Learning

In addition to evaluations on the learned stoichiometry representations, we also compare the models' performance in transfer learning scenarios in Table 4, where the encoder parameters are fine-tuned along with the MLP head. We have the following observations: **1)** Although the overall performance enhancement is observed due to the additional training of the encoder when compared with the results reported in Table 1, we sometimes observe that negative transfer occurs when comparing the Rand init. model and baseline methods in Table 4. This indicates that without an elaborate design of the tasks, pre-training may incur negative knowledge transfer to the downstream tasks [64]. **2)** However, by comparing to Rand init. in Table 4, we observe that PolySRL consistently leads to positive transfer to the model. We attribute this to the probabilistic representation, which maintains a high variance for uncertain materials, thereby preventing the representations of the materials from overfitting to the pretraining task.

Given that the primary objective of this paper is to propose a training approach for stoichiometry representation learning rather than introducing a new model architecture, previous supervised learning methods, i.e., **Roost** [20] and **CrabNet** [56], are not directly relevant to our research. Nevertheless,

we include a comparison with these previous works in this section to offer additional insights into our model's performance. For the experiment, we used publicly available codes provided by the authors [6] [7]. In Table 4, we observe that our simple stoichiometry encoder composed of GCNs and Jumping Knowledge Network (i.e., Rand init.) exhibits comparable or superior performance compared to the previous works that are elaborately designed for supervised stoichiometry learning. While previous works are elaborately designed for predicting properties of stoichiometry, they train the models from large-scale DFT-calculated properties of lowest enthalpy structures, giving up polymorphic structures of a single stoichiometry. However, in real-world scenarios, large-scale wet-lab experimental data is seldom available, which restricts their utility in the materials discovery process.

Table 4: Transfer learning performance including supervised learning baselines (MAE).

| Model | Band G. | Form. E. | Metallic | ESTM 300K | | | ESTM 600K | | |
| --- | --- | --- | --- | --- | --- | --- | --- | --- | --- |
| | | | | E.C. | T.C. | Seebeck | E.C. | T.C. | Seebeck |
| **Supervised Learning** | | | | | | | | | |
| Rand init. | 0.390 | 0.599 | 0.204 | 0.849 | 0.202 | 0.425 | 0.659 | 0.209 | 0.402 |
| | (0.012) | (0.053) | (0.014) | (0.174) | (0.027) | (0.048) | (0.098) | (0.019) | (0.082) |
| Roost | 0.384 | 0.743 | 0.199 | 0.851 | 0.216 | 0.406 | 0.684 | 0.240 | 0.402 |
| | (0.008) | (0.069) | (0.023) | (0.126) | (0.037) | (0.046) | (0.180) | (0.048) | (0.054) |
| CrabNet | 0.427 | 0.759 | 0.253 | 1.206 | 0.332 | 0.576 | 0.868 | 0.353 | 0.691 |
| | (0.012) | (0.052) | (0.023) | (0.042) | (0.046) | (0.080) | (0.138) | (0.040) | (0.057) |
| **Transfer Learning** | | | | | | | | | |
| GraphCL | 0.391 | 0.607 | 0.193 | 0.862 | 0.198 | 0.412 | 0.643 | 0.205 | 0.412 |
| | (0.011) | (0.026) | (0.018) | (0.236) | (0.031) | (0.006) | (0.098) | (0.021) | (0.077) |
| MP Band G. | **0.382** | 0.604 | 0.193 | 0.829 | 0.210 | 0.405 | 0.632 | 0.197 | 0.402 |
| | (0.012) | (0.036) | (0.025) | (0.187) | (0.038) | (0.006) | (0.095) | (0.028) | (0.081) |
| MP Form. E. | 0.391 | 0.582 | 0.197 | 0.822 | 0.195 | 0.410 | 0.641 | 0.209 | 0.428 |
| | (0.013) | (0.015) | (0.019) | (0.167) | (0.031) | (0.041) | (0.102) | (0.043) | (0.086) |
| 3D Infomax | 0.391 | 0.606 | 0.194 | 0.844 | 0.210 | 0.402 | 0.633 | 0.207 | 0.391 |
| | (0.006) | (0.027) | (0.019) | (0.195) | (0.032) | (0.005) | (0.133) | (0.018) | (0.077) |
| PolySRL | 0.386 | **0.576** | **0.191** | **0.822** | **0.189** | **0.386** | **0.626** | **0.195** | **0.390** |
| | (0.021) | (0.042) | (0.024) | (0.162) | (0.037) | (0.069) | (0.161) | (0.015) | (0.077) |

## F.3 Physical Validity

**Further Analysis.** In this section, we delve deeper into the physical validity of predicted properties for thermoelectrical materials by observing scatter plots that compare the actual ground truth values of $Z\bar{T}$ with the values obtained by the model predictions. For clearer visualization, we select one baseline model from models that consider DFT-calculated properties (i.e., MP Band G.) and structures (i.e., 3D Infomax). In Figure 6, we notice that the predictions produced by PolySRL consistently yield accurate calculations of $Z\bar{T}$ without any outliers. This observation underscores the model's ability to predict physically valid properties for thermoelectrical materials. Additionally, we observe that the model, specifically MP Band G., which lacks consideration of the structural information within stoichiometry, tends to produce outliers more frequently when contrasted with models that incorporate structural information. More specifically, three outliers made by MP Band G. in Figure 6 (a) are $Co_9S_8$, $Cu_5Sn_2S_{6.65}Cl_{0.35}$, and $Cu_{5.133}Sn_{1.866}S_{6.65}Cl_{0.35}$. In case of $Co_9S_8$, there exist only one possible structure in MP dataset, and there was no existing structure for $Cu_5Sn_2S_{6.65}Cl_{0.35}$, and $Cu_{5.133}Sn_{1.866}S_{6.65}Cl_{0.35}$. This suggests that MP Band G. encounters difficulty in acquiring accurate physical properties for materials where obtaining structural information is computationally challenging. On the other hand, in Figure 6 (b), two outliers made by MP Band G. are GeTe and SnTe, each of which has three possible structures in MP dataset. This indicates that MP Band G. suffers from obtaining valid physical properties from polymorphic structures. In conclusion, we argue that this finding underscores the significance of incorporating structural information for accurate predictions.

**High Throughput Screening.** As described in the main manuscript, the figure of merit $Z\bar{T}$ determines how effectively power can be generated and energy can be harvested across various real-world

---

[6]**Roost**: `https://zenodo.org/record/4133793`
[7]**CrabNet**: `https://github.com/anthony-wang/CrabNet`

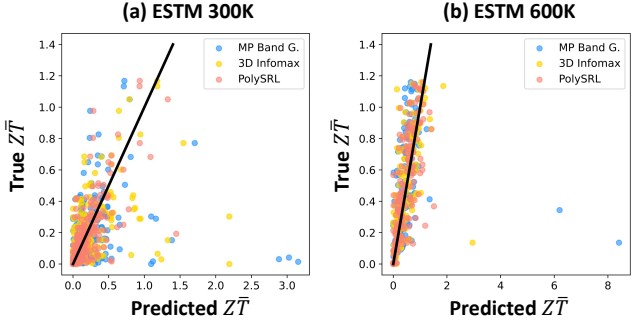

Figure 6: Scatter plot between true and predicted $Z\bar{T}$.

applications. To discover novel materials of high $Z\bar{T}$, we perform high-throughput screening based on the predicted $Z\bar{T}$ in Figure 7. In particular, for thermoelectrical materials at room temperature (300 K), we establish a threshold of $Z\bar{T} = 0.8$, and for high-temperature scenarios (600 K), we use a threshold of $Z\bar{T} = 1.1$. We observe that PolySRL outperforms all other baseline methods in ESTM 300K datasets, while performing competitively with 3D Infomax in ESTM 600K. This again demonstrates the importance of structural information in stoichiometry representation learning, which has been overlooked in previous works [20, 56].

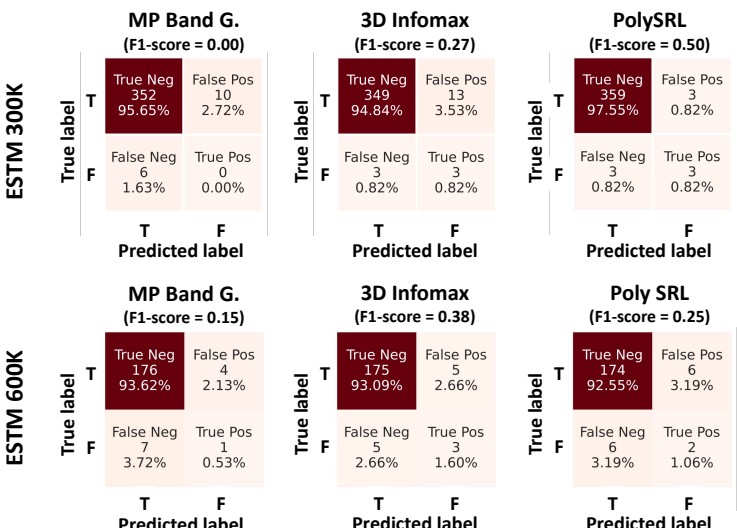

Figure 7: High throughput screening results.

## F.4 Model Analysis

**Ablation Studies.** In this section, we conduct ablation studies on our model by removing the sampling process described in Equation 3, which is denoted as "w/o Sampling" in Table 5. To clarify, rather than utilizing the sampled representations $\hat{\mathbf{z}}_j^a$ in Equation 4, we directly employ the mean vector of stoichiometry, denoted as $\mathbf{z}_\mu^a$, for the soft contrastive loss. By doing so, the model transitions from learning a probabilistic representation of stoichiometry to learning a deterministic representation of stoichiometry. To compare with methods that don't incorporate polymorphic structural information, such as 3D Infomax, we also present the performance of 3D Infomax in Table 5. We have the following observations: **1)** Considering polymorphic structure is crucial in stoichiometry representation learning by comparing 3D Infomax and w/o Sampling. **2)** Additionally, the sampling process typically leads to improved performance, underscoring the advantage of learning a probabilistic representation of stoichiometry. While w/o Sampling outperforms PolySRL in two datasets, the absence of the sampling process means the model can no longer estimate uncertainty in stoichiometry, thereby losing its practicality in real-world materials discovery. In summary, we argue

that PolySRL learns a probabilistic stoichiometry representation, which not only enables accurate uncertainty estimation but also enhances model performance.

Table 5: Ablation studies in representation learning scenarios (MAE).

| Model | DFT | | Poly. | Band G. | Form. E. | Metallic | ESTM 300K | | | ESTM 600K | | |
| | Prop. | Str. | | | | | E.C. | T.C. | Seebeck | E.C. | T.C. | Seebeck |
|---|---|---|---|---|---|---|---|---|---|---|---|---|
| 3D Infomax | ✓ | ✓ | ✗ | 0.428 (0.015) | 0.654 (0.032) | 0.201 (0.032) | 0.969 (0.110) | 0.217 (0.040) | 0.432 (0.070) | 0.692 (0.102) | 0.212 (0.013) | 0.428 (0.076) |
| w/o Sampling | ✓ | ✓ | ✓ | 0.410 (0.006) | 0.618 (0.060) | 0.198 (0.030) | **0.864** (0.192) | 0.208 (0.027) | 0.407 (0.054) | 0.679 (0.084) | 0.198 (0.011) | **0.396** (0.033) |
| PolySRL | ✓ | ✓ | ✓ | **0.407** (0.013) | **0.592** (0.039) | **0.194** (0.017) | 0.912 (0.121) | **0.197** (0.020) | **0.388** (0.059) | **0.665** (0.126) | **0.189** (0.017) | 0.412 (0.043) |

**Sensitivity Analysis on $\beta$.** In this section, we verify the empirical effect of the hyperparameter $\beta$, which controls the weight of the KL divergence loss computed between the learned distributions and the standard normal distribution, in Equation 6. We have the following observations from Figure 8 (a): **1)** As the hyperparameter $\beta$ increases, the average variance of the learned distributions (i.e., uncertainty) also increases, and the dimension of the variance vectors that collapse to zero (i.e., collapsed ratio) decreases. This indicates that the KL divergence loss effectively prevents the distributions from collapsing. **2)** On the other hand, the performance of PolySRL deteriorates as $\beta$ increases, indicating that emphasizing the KL divergence loss too much causes PolySRL to struggle in learning high-quality stoichiometry representations. However, reducing $\beta$ does not always result in improved performance, as collapsed distribution may not effectively capture information from polymorphic structures. Hence, selecting an appropriate value of $\beta$ is vital for learning high-quality stoichiometry representations while maintaining a suitable level of uncertainty.

**Sensitivity Analysis on Various Hyperparameters.** In addition, we provide an analysis on various hyperparameters in PolySRL, i.e., initial values of $c, d$ and number of samples $J$ in Equation 5. We have the following observations: **1)** While we made $c$ and $d$ learnable parameters to allow the model to adjust them adaptively to an optimal point, we've also found that setting the initial values for $c$ and $d$ is crucial in model training. This indicates that initial value plays a significant role in guiding the model correctly from the outset of the training process, ultimately contributing to good performance. **2)** On the other hand, we observe PolySRL shows robustness in various numbers of samples, suggesting that it can be trained effectively without a large number of samples, which will demand an extensive amount of computational resources.

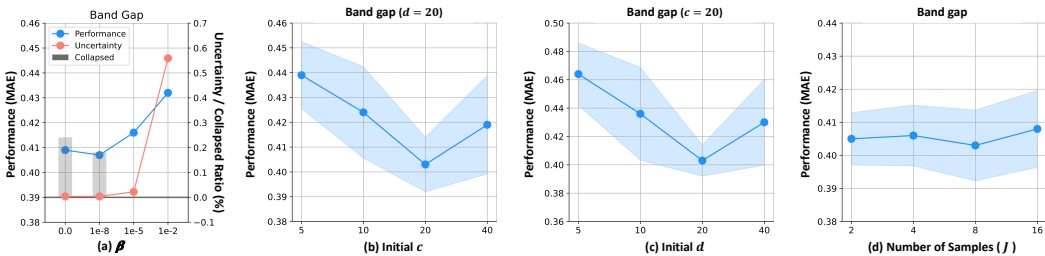

Figure 8: Additional sensitivity analysis results.

### F.5 Additional Uncertainty Analysis

**Uncertainty and Model Performance.** In this section, we analyze how the model performance varies regarding the uncertainties of the stoichiometry. To achieve this, we initially categorize stoichiometry based on MAE into intervals such as 0.0 to 1.0, 1.0 to 2.0, · · ·, and 4.0 to 5.0. For example, **Group 1** in Figure 9 (a) contains the group of MAE in the range 0.0 to 1.0. We then calculate the average uncertainties of the model for each group. As observed in Figure 9 (a), as the MAE values increase, the level of uncertainty also increases, demonstrating that the model effectively estimates uncertainties associated with MAE values.

**Additional Case Studies: Low Uncertainty with Multiple Structures.** In addition to the case studies in Section 5.3, we further provide cases where the stoichiometry with multiple possible

structures exhibits low uncertainty. In Figure 9 (b), we observe two stoichiometries with collapsed uncertainty, even though they possess four distinct possible structures. This phenomenon occurs because these structures share highly similar polymorphic arrangements, with only one unique structure in each stoichiometry. For instance, ZrC and $NdF_2$ predominantly adopt cubic and hexagonal structures, respectively, with only one distinct possible structure for each stoichiometry.

**Additional Case Studies: High Uncertainty with Multiple Structures.** In this section, we present additional case studies that align with our expectations. Figure 9 (c) illustrates two stoichiometries with the highest uncertainty among those possessing three polymorphic structures. For example, NaI can exist in three distinct structures (i.e., cubic, orthorhombic, and tetragonal), and AlP also exhibits three different structures (i.e., cubic, hexagonal, and tetragonal). Given that varying atomic arrangements within materials lead to entirely distinct physical and chemical properties, it becomes crucial to convey the extent of structural diversity that stoichiometry can exhibit during the material discovery process. Therefore, these additional case studies highlight the practicality of PolySRL in real-world material discovery.

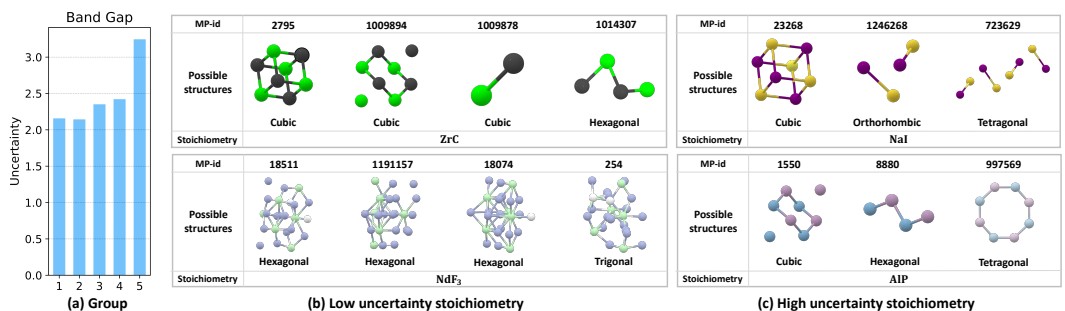

Figure 9: Additional uncertainty analysis.

# G Notations

In Table 6, we provide mathematical notations that are used in the main manuscript.

Table 6: Mathematical notations.

| Notations | Explanations |
|---|---|
| $n_s$ | Number of atoms in crystal structure |
| $\mathbf{X}^b$ | An elemental feature matrix of structural graph |
| $\mathbf{A}^b$ | An adjacency matrix of structural graph |
| $\mathcal{G}^b = (\mathbf{X}^b, \mathbf{A}^b)$ | A crystal structural graph |
| $\mathbf{z}^b$ | A latent representation of a crystal structural graph |
| $f^b$ | A GNN-based crystal structural encoder |
| $n_e$ | Number of unique elements in a stoichiometry |
| $\mathcal{E} = \{e_1, \ldots, e_{n_e}\}$ | A unique set of elements in a stoichiometry |
| $\mathcal{R} = \{r_1, \ldots, r_{n_e}\}$ | A compositional ratio of each element in a stoichiometry |
| $\mathcal{G}^a = (\mathcal{E}, \mathcal{R}, \mathbf{A}^a)$ | A fully-connected stoichiometry graph |
| $\mathbf{X}^a$ | A elemental feature matrix of stoichiometry graph |
| $\mathbf{A}^a$ | An adjacency matrix of stoichiometry graph |
| $\tilde{\mathbf{z}}^a$ | A sampled representation from latent distribution of stoichiometry |
| $f^a$ | A GNN-based stoichiometry graph encoder |
| $f_\mu^a$ | A mean module for stoichiometry graph |
| $f_\sigma^a$ | A variance module for stoichiometry graph |
| $J$ | Number of samples from latent distribution of stoichiometry (Equation 4) |
| $c$ | Learnable parameters for scaling the Euclidean distance (Equation 4) |
| $d$ | Learnable parameters for shifting the Euclidean distance (Equation 4) |
| $\mathcal{L}_{\text{con}}$ | Soft contrastive loss (Equation 5) |
| $\mathcal{L}_{\text{KL}}$ | KL divergence loss |
| $\beta$ | Hyperparameter that controls the weight of KL divergence loss |
| $\mathcal{L}_{\text{total}}$ | Total loss function (Equation 6) |

