# OpenReview forum: "Stoichiometry Representation Learning with Polymorphic Crystal Structures"
_NeurIPS.cc/2023/Workshop/AI4Science — NeurIPS2023-AI4Science Poster_

### Official Review · Reviewer_nVZZ · 2023-10-24
**Stoichiometry representation learning including information from crystal polymorphs, good work from AI perspective, lack of significance in science domain**

**Rating:** 9
**Confidence:** 3

**Review:**

The authors propose a model PolySRL that learns representations of crystals by simultaneously considering the stoichiometry and the crystal structural information, with the novelty of incorporating the polymorphism. The authors conduct extensive empirical studies on sixteen datasets, including wet-lab experimental data and DFT-calculated data, and demonstrate that PolySRL is quite effective in learning stoichiometry representations.

The authors provide comprehensive background information by summarising related work. Sufficient detail such that the proposed method appears reproducible. The paper reads quite well from AI perspective, and I suggest accepting it.

However, in chemistry, one expectation is using AI to predict new materials that outperform the current ones. How does PolySRL perform in this task? Another expectation involves analysing crystals or advancing our current understanding of the connection between crystal chemistry and their performance in specific applications. The model will have a big influence in crystal chemistry if the authors can demonstrate any of these abilities.

---

### Official Review · Reviewer_25y9 · 2023-10-24
**Review of 'PolySRL: Learning Probabilistic Stoichiometry Representations with Polymorphic Crystal Structures**

**Rating:** 7
**Confidence:** 4

**Review:**

This paper presents PolySRL, a novel approach for learning probabilistic representations of stoichiometry in crystalline materials, which incorporates polymorphic structural information to better capture the complexities in materials science. The paper conducts extensive experiments on sixteen datasets, including wet-lab experimental and DFT-calculated data, to demonstrate the effectiveness of PolySRL in learning stoichiometry representations. The approach not only outperforms baseline models but also highlights the model's ability to handle uncertainties stemming from polymorphism and computational feasibility in the material discovery process. It also conducts comprehensive uncertainty analysis to shed light on the model's behavior regarding polymorphism and impurities in materials.

Strong Points:
1. Innovative Approach: The paper addresses an important challenge in materials science by proposing a novel approach, PolySRL, which extends stoichiometry representation learning to incorporate polymorphic structural information, a key aspect of materials science that has been overlooked in previous works.

2. Extensive Experiments: The paper supports its claims with a rigorous evaluation process, conducting experiments on a wide range of datasets, including wet-lab experimental and DFT-calculated data, to showcase the superiority of PolySRL over baseline methods.

3. Uncertainty Analysis: The paper's in-depth analysis of uncertainties provides valuable insights into the model's performance and highlights its ability to capture various complexities in materials science, making it a practical tool for real-world material discovery.

Weak Points:
1. Complexity: The paper introduces a complex model with multiple components, which may make it challenging for readers without a strong background in machine learning or materials science to follow and implement.

2. Practical Applicability: While the paper shows the model's effectiveness in capturing uncertainties, it would be beneficial to include real-world use cases or applications where these uncertainties are of practical importance in materials science.